# The Rift Valley fever (RVF) vaccine candidate 40Fp8 shows an extreme attenuation in IFNARKO mice following intranasal inoculation

**Belén Borrego**[1]*, **Celia Alonso**[1], **Sandra Moreno**[1], **Nuria de la Losa**[1], **Pedro José Sánchez-Cordón**[2], **Alejandro Brun**[1]*

**1** Department of IMMUNOLOGY, PATHOLOGY AND CONTROL OF INFECTIOUS DISEASES, Centro de Investigación en Sanidad Animal CISA INIA/CSIC, Valdeolmos, Madrid, Spain, **2** Department of INFECTIOUS DISEASES AND GLOBAL HEALTH, Centro de Investigación en Sanidad Animal CISA INIA/ CSIC, Valdeolmos, Madrid, Spain

* borrego@inia.csic.es (BB); brun@inia.csic.es (AB)

## Abstract

Rift Valley fever (RVF) is an important zoonotic viral disease affecting several species of domestic and wild ruminants, causing major economic losses and dozens of human deaths in various geographical areas of Africa, where it is endemic. Although it is not present in Europe, there is a risk of its introduction and spread linked to globalisation and climate change. At present, the only measure that could help to prevent the disease is vaccination of flocks in areas at risk of RVF. Available live attenuated vaccines are an effective means of controlling the disease, but their use is often questioned due to residual virulence, particularly in susceptible hosts such as pregnant sheep. On the other hand, no vaccine is currently licensed for use in humans. The development of safe and effective vaccines is therefore a major area of research. In previous studies, we selected under selective mutagenic pressure a highly attenuated RVFV 56/74 virus variant called 40Fp8. This virus showed an extremely attenuated phenotype in both wild-type and immunodeficient A129 (IFNARKO) mice, yet was still able to induce protective immunity after a single inoculation, thus supporting its use as a safe, live attenuated vaccine. To further investigate its safety, in this work we have analysed the attenuation level of 40Fp8 in immunosuppressed mice (A129) when administered by the intranasal route, and compared it with other attenuated RVF viruses that are the basis of vaccines in use or in development. Our results show that 40Fp8 has a much higher attenuated level than these other viruses and confirm its potential as a candidate for safe RVF vaccine development.

## Author summary

Vaccines that use attenuated viruses are highly effective in terms of providing protection, duration, breadth, and quality of the immune response. However, they may pose a risk to

---

**Funding:** Work supported by grants AGL2017-83326-R funded by Ministerio de Ciencia e Innovación MCIN/AEI/ 10.13039/501100011033 and by "ERDF A way of making Europe" and PdC2021-121717-I00 funded by MCIN/AEI/ 10.13039/501100011033 and by the "European Union Next Generation EU/PRTR". The funders had no role in study design, data collection and analysis, decision to publish, or preparation of the manuscript.

immunosuppressed individuals or in certain situations, such as during pregnancy. It is therefore important to analyze the residual virulence of attenuated vaccines to ensure their safety. In this work, we analysed the safety of the RVF virus vaccine variant 40Fp8 developed in our laboratory. We studied its effect on immunodeficient mice and used a more aggressive route of administration such as the intranasal delivery route. To assess the degree of attenuation of 40Fp8, we compared it with other attenuated prototype RVF vaccines. The results clearly demonstrate that 40Fp8 is highly attenuated, and its residual virulence level is extremely low compared to other vaccine viruses tested.

## Introduction

Rift Valley fever (RVF) is an infectious disease considered a major threat to animal health and the economy, due to the abortions and deaths it causes in domestic ruminants (mainly sheep, goat, cattle and camels). Host susceptibility varies with species and age. Sheep, and especially young animals with deaths up to 90–100%, are most susceptible. In adults, infection presents with fever and unspecific symptoms, except for the so-called abortion storms, stillbirths and fetal malformations, which are the most characteristic manifestations [1]. Because of its zoonotic nature, RVF constitutes also a huge threat to food security and public health services in developing countries. In humans, most infections are asymptomatic or appear as a mild influenza-like illness, although in some cases it can develop into a more severe haemorrhagic or neurological illness, which in a small percentage of cases can be fatal [2–5].

The disease is caused by the Rift Valley fever virus (RVFV), a Bunyavirus within the Phlebovirus genus of the Phenuiviridae family. The virus has a trisegmented RNA genome, with the large (L) segment coding for the viral polymerase, the Medium (M) segment with coding capacity for two mature glycoproteins (Gn and Gc) as well as the non-structural NSm and 78kDa proteins, and the Small (S) segment coding for the nucleocapsid protein (N) and a non-structural protein (NSs). NSs is considered the main virulence factor because of its ability to inhibit the host innate immune system by multiple pathways that, either alone or combined, allow the virus to replicate efficiently [6].

RVFV is transmitted by mosquitoes, and humans can also be infected when exposed to blood and other tissues of sick animals, through inhalation of aerosols or by consuming raw products. Since its first description in 1931, RVFV epizootics and epidemics have continuously occurred in the African continent, mainly in sub-Saharan countries and Egypt, and since 2000 also in the Arabian Peninsula and several islands off the coast of southern Africa. Recent data show that the virus has spread to northern African countries, with active outbreaks and detection of seropositive animals [7] (PROMED & WHOAFRO 2022, 2023). To date no cases have been reported in Asia, Europe or the Americas, with the exception of a few human imported cases [8]. However, the mosquitoes associated with transmission of the virus are widespread worldwide, highlighting the risks of disease spread associated with climate change and globalisation [9].

Together with ecological measures focused on the control of insect populations, vaccination is considered the most efficient way to control the disease. Live attenuated vaccines as well as inactivated vaccines are commercially available in Africa for veterinary use. Inactivated vaccines have some limitations since repeated doses are needed for a good immunisation, whereas vaccines based on live attenuated viruses (LAV) rapidly induce long-lasting and broadly protective immunity after a single inoculation, making them an excellent choice for immunisation programmes. However, there is some concern about the safety of these vaccines due to the

unlikely, but possible, genetic reassortment between closely related virus strains in endemic areas, and especially due to the possibility of reversion and/or the presence of residual virulence. These facts tend to limit the use of live attenuated vaccines in immunocompromised hosts or pregnant domestic ruminants and highlight the need for novel, safer efficient LAV vaccines (reviewed in [10, 11]).

There are two main prototypes of attenuated RVF vaccines based on their mechanism of attenuation. One is based on the lack of NSs function, as in the absence of this protein the virus is unable to counteract the innate response of immunocompetent hosts, resulting in an attenuated virus. This was first observed with the Clone 13 strain, a plaque isolate virus obtained from a non-fatal human case in the Central African Republic, which carries a natural deletion of 69% of the NSs gene [12, 13]. Vaccines based on Clone 13 are commercially available and widely used in several African countries. However, other virulence determinants remain, since an overdose in pregnant sheep lead to malformations and stillbirths [14], and in mouse models, mutant viruses obtained by deletion of NSs retained high virulence when inoculated intranasally [15, 16].

The second attenuation mechanism, exemplified by the MP-12 strain, consists of the combination of several single changes located on the different genomic segments. MP-12 is derived from the pathogenic Egyptian human strain ZH548, and was obtained by serial passaging in cell culture in the presence of a mutagenic agent [17,18]. MP-12 has been extensively characterised and shown to be a safe and effective immunogen in both animals and humans (reviewed in [19,20]). It has been tested in clinical trials and proposed as a human vaccine candidate with a conditional licence for veterinary use in the USA [21–23], and further modified to improve its vaccine potential. However, MP-12 is fully pathogenic for certain strains of immunodeficient mice [13,24,25], and a few studies have reported mild liver damage in lambs or calves [26,27] and abortions and teratogenic effects in pregnant sheep vaccinated from 35 to 56 days of gestation [28,29].

In previous work by our group, a new variant of RVFV was obtained by serial passaging of the South African RVFV 56/74 in cell culture in the presence of favipiravir, an antiviral drug with mutagenic activity [30]. This virus, designated 40Fp8, was proposed as a safe vaccine candidate because it showed an attenuated phenotype in wild-type (wt) mice while retaining its ability to induce a protective immune response. Furthermore, 40Fp8 was found to be highly attenuated also in immunodeficient A129 (interferon-alpha receptor gene knock-out) mice when inoculated intraperitoneally [31], a finding of particular relevance given the high susceptibility of these mice to viral infections [32,33].

In this work we analysed the degree of attenuation exhibited by 40Fp8 when inoculated into A129 mice intranasally, a more severe route of infection that leads to high mortality rates even in immunocompetent animals [16]. The two aforementioned viruses considered as prototypes of attenuated vaccines, MP-12 and a NSs- deleted virus expressing GFP instead, short named rG1, were included for a comparison. While all mice inoculated with these vaccine viruses succumbed in the very first days after infection, those inoculated with our RVFV vaccine candidate 40Fp8 showed survival rates close to 100%, confirming the hyperattenuation of this virus and thus its potential as a extremely safe vaccine candidate.

## Materials and methods

### Ethics statement

All experimental procedures were performed in accordance with EU guidelines (directive 2010/63/EU), and protocols approved by the Animal Care and Biosafety Ethics' Committees of

Centro Nacional Instituto Nacional de Investigación y Tecnología Agraria y Alimentaria (INIA-CSIC) and Comunidad de Madrid (Authorization decision PROEX 079.6/22).

## Viruses and cells

The three viruses analyzed in this work are 40Fp8, a favipiravir mutagenized variant derived from the RVFV South African isolate 56/74 [30, 34]; rG1, a recombinant NSs-deleted RVF virus expressing the green fluorescent protein, GFP, that was previously obtained [31,35], and rMP-12, the RVFV vaccine strain MP-12 recovered from cDNAs, kindly provided by Dr. Makino, UTMB, Galveston, USA [36]. Viruses were grown and titrated in Vero cells (African green monkey kidney cells, ATCC CCL-81) following standard procedures as described [34]. Inocula corresponded to rescued viruses after 3 (rG1) or 5 (rMP-12) passages in cell culture. For 40Fp8, a viral stock was obtained from the original isolate by inoculation of Vero cells at a moi of 0.2 and its genetic stability was confirmed by Sanger and NGS sequencing [37].

## Mice inoculation and sampling

*In vivo* studies were done using A129 (interferon-alpha receptor gene knock-out, IFNARKO) mice bred in our own facilities at the Department of Animal Reproduction (INIA/CSIC); mice were housed in biosafety level 3 (BSL-3) animal room at CISA before use and distributed into experimental groups of 8 individuals (4 males and 4 females). Mice were intranasally inoculated with the corresponding viruses at the indicated doses in a volume of 10 microliters per nostril under general anesthesia. Weight was controlled daily over 2 weeks (circa), as well as development of disease in terms of morbidity (watery eye, ruffled hair, hunching, activity) and mortality. Animals were euthanized when showing severe signs of disease (lethargy, ataxia, tremors) according to a humane endpoint score. Blood samples were taken by submandibular puncture at 72 h after infection to monitor viremia, while serum samples to be used in antibody assays were collected at the end of the experiment [31].

For some selected animals brain and liver were aseptically removed and samples directly frozen at -80˚C or kept in RNAlater solution (Ambion) at -20˚C until processing. Besides, different tissue samples (liver, brain, spleen and kidney) were fixed by immersion in 4% buffered formalin solution for 72 hours. Then, samples were routinely processed and embedded in paraffin wax for subsequent histopathological and immunohistochemical studies.

## RNA extraction and viral RNA detection

3 dpi blood samples were processed for RNA extraction as described [31] using the Speedtools RNA virus extraction kit (Biotools B&M Labs, S.A., Madrid, Spain). Brain and liver samples kept in RNA later were processed using the Promega SV total RNA isolation System kit. Briefly, after removal of RNA later, samples (30 mg of liver and 60 mg of brain) were washed and homogenized in 175 microliters of RNA Lysis Buffer with 1% beta-mercaptoethanol in 2 ml tubes with one 5-mm stainless steel homogenization bead. One homogenization cycle (50 Hz oscillation frequency (50 cycles/s) for 5 minutes) was run in a tissue Lyser LT (Quiagen). Then, homogenates were processed according to the manufacturer's protocol. RNAs extracted as described were collected in a final volume of 100 microliters, and 1 microliter was then tested for viral RNA detection by a real-time RT-qPCR specific to the RVFV L-segment [35,38].

## Tissue homogenization and infectious virus isolation

Liver and brain samples frozen at -80˚C were processed to obtain 10% weight/volume homogenates in DMEM, by running one homogenization cycle (50 Hz oscillation frequency (50

cycles/s) for 4 minutes) in a Tissue lyser LT (Qiagen). Tubes were then centrifuged at 5000 rpm for 5 min, supernatants collected and stored at -80˚C until use. To check for infectious virus, a volume of 5–20 microliters of 3 dpi-whole blood (depending on the availability) or 10 and 100 microliters of tissue homogenates (corresponding to 0.1 and 1 of tissue respectively) were inoculated onto Vero cell monolayers. After 1–2 hours of adsorption at 37˚C, inocula were removed, the monolayers extensively washed, and fresh medium was added. Cultures were kept in the incubator at 37˚ C and checked up to 6–7 days for cytopathic effect (cpe) (or GFP fluorescence in the case of virus rG1), and if negative, subjected to two further blind passages. Supernatants collected after the last one were analyzed by indirect immunofluorescence using a rabbit polyclonal anti-RVFV serum or the in-house monoclonal antibody 2B1[35], in order to confirm specificity of the infection in case of positive cpe, or to exclude infection in the absence of cpe. Some samples scored as positive were also titrated by a standard plaque assay on Vero cells.

## Antibody assays

RVFV neutralizing antibodies were detected in a microneutralization assay in MW96 plates and antibodies against NP by an in-house ELISA as described [31]. Neutralization titers are expressed as the dilution of serum (log2) rendering a reduction of infectivity of 50%, while anti-NP titers are expressed as the OD reading at 450 nm at a fixed dilution of serum.

## Histopathological and immunohistochemical studies

From the paraffin blocks containing the tissue samples, serial sections 4 μm thick were obtained using a rotary microtome. These sections were used for routine histological staining (haematoxylin-eosin) for histopathological evaluations.

To visualise the viral antigen, immunohistochemistry was performed on tissue sections following previous protocols [37] which were adapted to mouse tissues. Briefly, after inhibition of endogenous peroxidase activity, samples were subjected to heat-induced epitope retrieval by immersion in citrate buffer (pH 6; 0.01M) at sub-boiling for 10 min. After rinsing in TBS buffer (pH 7.2), non-specific binding sites were blocked with 20% normal goat serum in TBS buffer for 30 min. The sections were then incubated with an in-house polyclonal rabbit anti-RVFV serum (diluted 1:1200 in TBS buffer) overnight at 4˚C. The following steps were as described [37]. Tissue samples from infected and uninfected mice were included as test controls. In addition, an in-house irrelevant rabbit antiserum was used as as a negative technical control for the primary antibody. Cells immunolabelled for viral antigen were semi-quantitatively assessed in tissue sections as follows: (0) absence of immunolabelled cells; (1) occasional presence of immunolabelled cells; (2) mild presence of immunolabelled cells; (3) moderate presence of immunolabelled cells; (4) abundant presence of immunolabelled cells. Morphological characteristics, location and size were the criteria applied to identify immunolabelled cell types.

## Statistical analysis

Due to the exploratory nature of this research work, a formal hypothesis on the expected outcomes was not pre-specified. To control variation, inbred animals were used, which allows to reduce the number of animals per experimental unit using a sample size of n = 8. This sample size has a 95% power to detect an effect size of 12.5% in means, assuming equal variances, and an effect size of 0.7 in frequencies, for a 2% significance level and a two-sided test [39]. Mice were randomly distributed with 50% males and females per group. Investigators were not blinded to group's outcome assessment. All statistical evaluations were performed using the

GraphPad Prism version 8.0 (GraphPad Software, San Diego, CA). Survival curves were compared using the log rank (Mantel-Cox) test. Comparison of viral RNA qRT-PCR and immune response data between groups was performed using multiple t-tests (Holm-Sidak method and Mann-Whitney test respectively). Differences were considered statistically significant when $p < 0.05$.

## Results

### Virulence assesment of attenuated RVFVs in IFNARKO mice

A129 IFNARKO mice provide a highly sensitive system for assessing the degree of viral attenuation [32,33]. On the other hand, intranasal (IN) inoculation of RVFV constitutes an extremely severe route of infection, with immunocompetent mice succumbing after inoculation with attenuated viruses [16]. Combining these approaches, i.e., IN inoculation of A129 mice, we compared the infectivity of the 40Fp8-RVFV isolate with two viruses representing the prototypes of live attenuated vaccines. These viruses were rMP-12, whose attenuation is achieved by a combination of several single attenuation determinants [18], and rZH548ΔNSs::GFP, short-named as rG1, a recombinant virus lacking the NSs protein which is considered to be the main virulence factor of RVFV [13,35].

8–12 week old A129 mice distributed in groups of 4 males and 4 females were IN inoculated with 2 doses (low and high, 50 and 1000 pfu, respectively) of each one of the three viruses. Animals were monitored daily for 2 weeks to assess weight variation, signs of illness and survival; on day 3 blood samples were taken for analysis of viremia by RT-qPCR. Those that succumbed or were euthanised were selected for histopathological (HP) and immunohistochemical (IHC) examination. On day 15 the experiment was terminated and survivors were euthanised and sampled to assess the development of antibodies against viral proteins and/or for HP and IHC analyses (S1 Fig).

Survival curves showed the extreme attenuation of 40Fp8, with survival rates close to 100% (Fig 1A). Only one animal (out of 8) in the group receiving the high dose showed paralysis and ataxia on day 9 pi and was therefore euthanised and sampled for HP and IHC studies (mouse F_H_4). In addition to this high survival rate, animals inoculated with 40Fp8 remained healthy throughout the experiment, except for 4 mice in the high dose group (M_H_2, M_H_3, F_H_1 and F_H_3, S1A Table), that showed some weight loss and/or mild clinical signs (watery eye) between days 8 and 10 pi (Fig 1B and 1C). They all regained normal weight by the end of the experiment; nevertheless they were still selected for HP and IHC analysis when sacrificed at the end of the experiment (day 15 pi).

Conversely, the groups receiving the other LAV prototype viruses, rG1 and rMP-12, produced no survivors, even those inoculated with the low virus dose. In these groups all mice became ill and succumbed very quickly, between days 3 and 5; the best median survival time (MST) was 4.5 days in the group inoculated with 50 pfu of rMP-12.

HP and IHC studies showed a very similar pattern for infection with both LAV prototype viruses, rG1 and rMP-12, regardless of the dose of virus received or the sex of the animal. As previously suggested by the MST, infection with rMP-12 appeared to be slightly milder, with later deaths and slightly lower viral antigen detection scores (Fig 2). However, histopathological lesion patterns, virus target cells and viral antigen distribution in the organs evaluated were similar in both experimental groups, with massive presence of viral antigen in the main target organs (liver and spleen), together with the occasional presence of immunolabelled cells in the kidney (Fig 2B–2F). In rG1-inoculated animals that died later (d4, d5), virus was also detected in brain samples but associated with circulating mononuclear cells (Fig 2G).

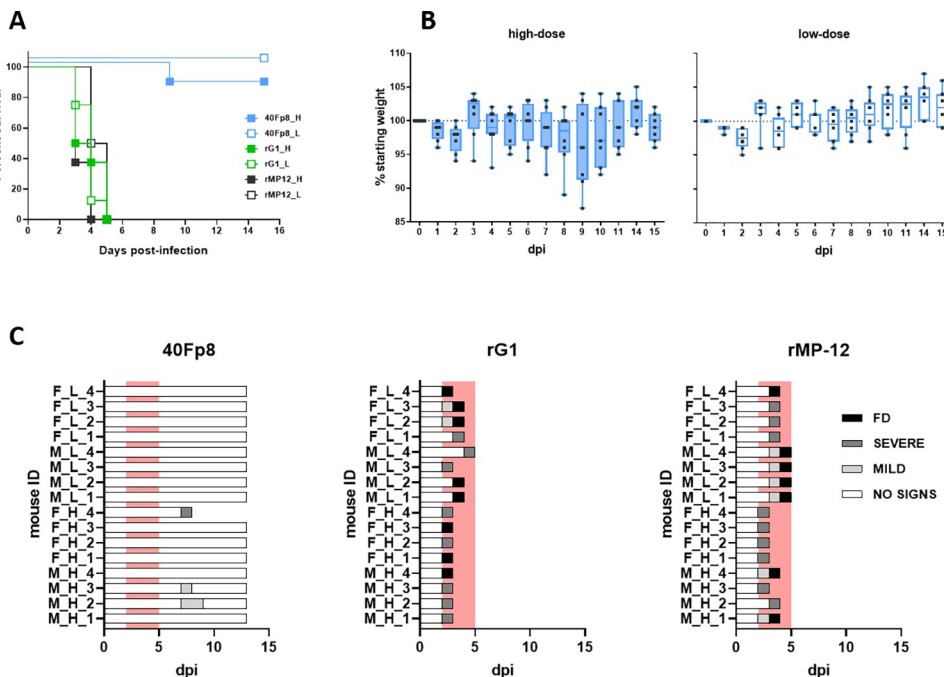

**Fig 1. Intranasal inoculation of A129 mice with attenuated viruses.** Groups of n = 8 A129 mice 8–12 weeks old were inoculated by the IN route with a low (L) or high (H) dose (50 or 1000 pfu, respectively, full and empty symbols) of the indicated viruses: 40Fp8 (blue), rG1 (green) and rMP-12 (black). Each group included 4 Female (F) and 4 Male (M). Doses were confirmed by titration of the inocula. Animals were monitored daily to check for signs of disease. (**a**) **Survival curves.** Percentages of survival are represented using Kaplan-Meier plots. Survival distributions were compared by Log-rank (Mantel-Cox) test. The survival curves are significantly different with p <0.0001. (**b**) **Weight variation.** For each individual mouse, weight at the beginning of the experiment was considered as 100% (thick line) and used to calculate the percentage of variation along the experiment. Results of the group are represented in the "box and whiskers" format, with individual values as black points, and the median plotted as a line in the middle of the box. Results are only shown for the groups inoculated with 40Fp8. Left panel (full boxes), animals receiving the high dose; right panel (empty boxes), animals receiving the low dose. (**c**) **Clinical signs.** Signs observed were classified as mild (ocular watery discharge and/or ruffled hair), or severe (strongly reduced activity and/or paralysis and/or tremors) that led to euthanasia of the animal. FD = found dead. The area corresponding to days 2 to 5 pi has been red-shaded to highlight the period when deaths occured in the rG1 and rMP-12 groups.

On the other hand, in the 40Fp8-inoculated mouse sacrificed on day 9 (F_H_4), no viral antigen was detected in any of the samples analysed, including the main target organs such as the liver (Fig 2H). Histopathological evaluations showed the presence of moderate to severe multifocal meningitis, although no cells immunolabelled against the viral antigen (neither circulating nor belonging to the central nervous sytem cells) were observed (Fig 2I). The tissue samples taken from survivors selected at the end of the experiment as having shown mild clinical signs or weight loss throughout the experiment were completely negative for viral antigen detection (S2 Fig).

RT-qPCR analysis of viral RNA load in blood samples collected at d3 pi revealed clear differences between the groups under study (Fig 3A). Animals receiving 40Fp8 (both high and low doses, blue symbols) were considered negative, with Cq values very close or even below the detection level of the assay (Cq 37.0), and in the "doubtful"area delimited by negative samples (shaded in the figure). On the other hand, mice inoculated with rG1 (green symbols) showed mostly positive values, while those inoculated with rMP-12 (black) showed some differences according to the dose received, clearly positive in the high-dose group, and less so in the low-dose group, with an important number of samples in the doubtful area. Despite this result, these animals succumbed to the infection in the next days.

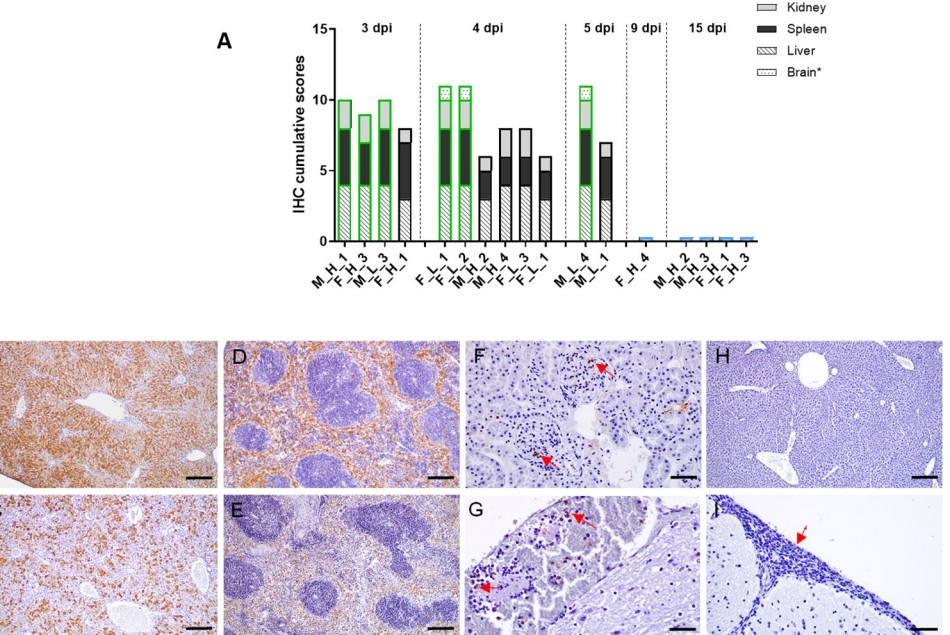

**Fig 2. Histopathological studies.** Animals succumbing or euthanized along the experiment at the days pi indicated were selected for histopathological (HP) and immunohistochemical (IHC) studies. Results shown correspond to selected/representative individuals within each virus group, identified by sex (M/F) and viral dose received (H/L) as in the legend of Fig 1. Data for each animal are shown in S1 Table. (**a**) **Individual representation of the cumulative scores** of cells immunolabelled against RVF virus antigen in tissue samples obtained on different days pi from selected mice within each virus group. Immunolabeled cell score (y-axis); Mice evaluated in each virus group (x-axis). The borders of the bars have been coloured according to the virus inoculated: green: rG1; black: rMP-12; blue: 40Fp8. For graphical purposes, animals inoculated with 40Fp8 have been assigned a value of 0.5 although all animals had a value of 0. (**b-i**) **Representative images of tissue sections immunolabelled against RVF virus antigen.** Massive presence of cells immunolabelled against RVF virus antigen in the liver (b, c) and spleen (d, e) of rG1 (b, d) and rMP-12 (c, e) inoculated mice. Note a slightly lower presence of viral antigen in the rMP-12-inoculated mouse (c, e). Observe also the occasional presence of immunolabelled cells in the kidney of rG1 (f; arrows) inoculated mice, as well as the presence of labelled circulating mononuclear cells within the meningeal blood vessels (g; arrows). Neither in the liver (h) nor in the brain (i) of the 40Fp8-inoculated mouse euthanised on day 9 pi were cells immunolabelled against the viral antigen detected. Note the severe menigitis consisting mainly of lymphocyte infiltrates (i; arrow). Liver (b) and spleen (d) of rG1(FD) F_H_3 mouse (3 dpi); Liver (c) and spleen (e) of rMP-12(EU) F_H_1 mouse (3 dpi); Kidney (f) of rG1(EU) M_L_3 mouse (3 dpi); Brain (g) of rG1(FD) F_L_2 mouse (4 dpi); Liver (h) and brain (i) of 40Fp8 (EU) F_H_4 mouse (9 dpi). IHC, scale bars: 50 micrometers (f, g, i); 200 micrometers (b, c, d, e, h).

In order to clarify this outcome, whole blood samples with Cq above or close to 33.0, i.e., in the doubtful range, were further tested on cultured cells for the detection of infectious virus. All samples collected from the 40Fp8-inoculated mice were negative after three blind passages. In contrast, samples from rG1 and rMP-12 inoculated mice were positive, most of them at the first passage. Only 1 sample from the group inoculated with the low dose of rMP-12 remained negative after 3 passages (mouse M_L_1, S1 Table).

To confirm that the surviving 40Fp8-inoculated animals had indeed supported some level of viral replication, we analysed seroconversion in serum samples collected at the end of the experiment by neutralisation and anti-N ELISA assays (Fig 3B and 3C). With the exception of one animal whose sample showed some neutralization capacity only at the first dilution tested (1/10), mice inoculated with the high dose showed high neutralizing titers even at the highest dilution tested (1/320). Antibody levels were lower in the group inoculated with 50 pfu, with only 2 animals clearly positive and the others close to the detection limit of the assay. When tested in the anti-N ELISA, all these low-dose samples gave values above the cut-off value of

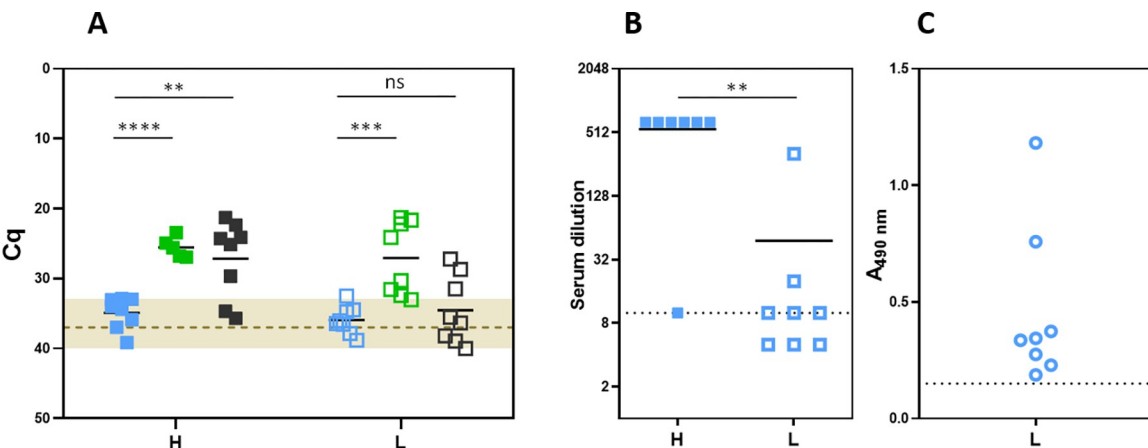

**Fig 3. Analysis of samples from IN inoculated mice. (a) Viral load** as determined by RT-qPCR in blood samples collected at day 3 pi from mice inoculated with a high (H, full symbols) or low (L, empty symbols) dose (1000 or 50 pfu, respectively) of virus 40Fp8 (blue), rG1 (green) or rMP-12 (black). Each point corresponds to an individual. Bars indicate the mean value. The area defined by negative control samples (negative controls in each round of PCR giving Cq values between 33 and 40) has been shaded. A line has been depicted at Cq = 37, considered arbitrarily as the detection level of the assay. Statistical significance determined by multiple t-tests (Holm-Sidak method). P values: 40Fp8 vs rG1: 0.000005 (high dose), 0.0005 (low dose); 40Fp8 vs rMP-12: 0.0025 (high dose), 0.4676 (low dose). **(b) Neutralizing antibodies in survivors inoculated with 40Fp8.** Serum dilution (log2) at which 50% of the wells were protected from cpe as observed in control wells. H (filled squares): animals inoculated with the high dose, i.e. 1000 pfu; L (empty squares): low dose, 50 pfu. Each point corresponds to an individual. Bars indicate the mean value. Samples were tested from 1/10 dilution (sensitivity limit of the assay, indicated by a dotted line) and then serially diluted 2-fold to 1/320. For the purpose of the graph, samples in which cpe was observed at the first dilution tested were considered negative and assigned an arbitrary value of 1/5 dilution; similarly, samples that still showed total neutralisation at the last dilution tested were not further tested and assigned a value of 1/640 dilution. Statistical significance determined by Mann-Whitney test. P value: 0.0023. **(c) AntiN ELISA** was performed only on samples from the low dose group (empty circles). Samples were tested in serial 3-fold dilutions from 1/50; graph shows the OD reading values given by samples diluted 1/150. The line corresponds to the cut-off value of 0.150, established as 2x the reading of the blank.

the assay, indicating seroconversion to this protein (i.e. some degree of infection) in all animals. These results confirm that, although no virus was detected or isolated from samples collected on day 3 pi, some level of replication did occur in animals within this group, leading to the development of antibodies against viral proteins, but this replication did not lead to disease or death in most animals.

## Pathological findings in scheduled euthanized animals

The above results demonstrated the extreme attenuation of 40Fp8 in terms of survival of IFNARKO mice even when inoculated intranasally with 1000 pfu. With the exception of one animal that was euthanised on day 9 with neurological signs, animals inoculated with the high dose remained healthy throughout the experiment, showing only mild signs of disease and some weight loss between days 8 to 10 pi (Fig 1B and 1C), and recovered from day 11 onwards.

In order to gain some insight into a (possible) pathogenicity of 40Fp8 that might have been overlooked in the design of the previous experiment, we decided to perform a new experiment in which animals inoculated with the high dose (1000 pfu) were scheduled to be euthanized at different times after infection and major target organs examined for viral antigen detection and/or tissue injury. Based on the results of the first experiment, the time points chosen were 2, 5 and 9 dpi. In the previous experiment, we collected samples on days 3, 4 and 5 pi, when the deaths occurred. For comparison among the three viruses under study, we chose day 2 as the last day with healthy animals. Day 9 was chosen as the time point when the highest weight loss was recorded and the only death occurred in the 40Fp8-inoculated group (Fig 1B and 1C).

**Table 1. RVFV detection in organs.**

| dpi | # mice ID | | LIVER | | BRAIN | |
| --- | --- | --- | --- | --- | --- | --- |
| | | | VI | RT-qPCR | VI | RT-qPCR |
| 2 | 40Fp8 | 1 | NEG | NEG | NEG | NEG |
| | | 2 | NEG | NEG | NEG | NEG |
| | rG1 | 1 | NEG | NEG | NEG | NEG |
| | | 2 | (++) | 31.26 | NEG | NEG |
| | rMP-12 | 1 | (++) | 30.18 | NEG | NEG |
| | | 2 | (+) | NEG | NEG | NEG |
| 5 | 40Fp8 | 1 | NEG | NEG | NEG | 34.29 |
| | | 2 | (++) | 32.31 | NEG | NEG |
| 9 | 40Fp8 | 1 | NEG | NEG | NEG | 36.29 |
| | | 2 | NEG | NEG | NEG | NEG |
| | | 3 | NEG | NEG | NEG | NEG |

11–15 week old A129 mice, including both males and females, were IN inoculated with 10 microliters of a viral suspension containing 1000 pfu of rG1, rMP-12 (n = 2) and 40Fp8 (n = 7). Animals were monitored for signs of disease and survival only and euthanized on days 2, 5 and 9. Based on previous survival results, groups receiving rG1 and rMP-12 were scheduled only for day 2. Liver and brain samples were collected and processed to test for viral load, both by infectious virus isolation (VI) on cell culture and by RT-qPCR. The results of virus isolation are expressed as follows: NEG (negative, no cpe); (+) (positive, cpe registered only when inoculated with the first dilution of the homogenate, corresponding to 1 mg of tissue), and (++) (strongly positive, when both dilutions, corresponding to 0.1 and 1 mg of tissue, yielded cpe). The specificity of the cpe was confirmed by IFI. For RT-qPCR results, the Cq given by the sample is indicated; those with a value greater than 37 are reported as Negative (NEG).

Day 5 was used as an intermediate time point, when the last deaths were observed in the rG1 and rMP-12 low dose groups (Fig 1A and 1C). Due to the rapid and significant mortality observed in animals inoculated with rG1 and rMP-12 viruses, these groups were only included for a comparison on day 2 pi. Mice were evaluated only for signs of disease and survival, and were euthanised on the specified days. Brain and liver samples were collected for virus isolation and RT-qPCR while liver, brain, spleen and kidney samples were fixed in 4% buffered formalin solution for HP and IHC studies. No signs of disease were observed in any animal throughout the experiment.

On day 2 pi, no virus was isolated from any of the samples taken from the 2 animals inoculated with 40Fp8, whereas infectious virus was recovered from liver samples from the other groups: 1 out of 2 within rG1, and both samples from rMP-12-inoculated mice, albeit with different viral loads (Table 1). When titrated in a plaque assay, that coming from G1#2 gave a titer of $10^2$ pfu/mg of tissue; the others were below the detection limit of the assay and viral loads were estimated to be below 10 pfu/mg. In all cases, positive samples were detected at the first passage in cell culture and RVFV-specific cpe was confirmed by IFI. Cpe was only observed in cultures inoculated with liver samples; none of the brain samples were positive.

These results were quite consistent with those obtained by HP and IHC analysis (Fig 4), with some cells immunolabelled against the viral antigen in liver and spleen samples from the indicated animals, although the number of immunostained cells was much lower than that observed on day 3 pi during the first experiment (Fig 2). Comparison with viral detection by RT-qPCR revealed a slightly lower sensitivity of this assay, with sample rMP-12#2 giving a negative Cq value, although cell culture had allowed recovery of infectious virus.

In samples collected at later times after infection from mice inoculated with 40Fp8, infectious virus was isolated from only one liver sample collected on day 5 (mouse #2). This sample

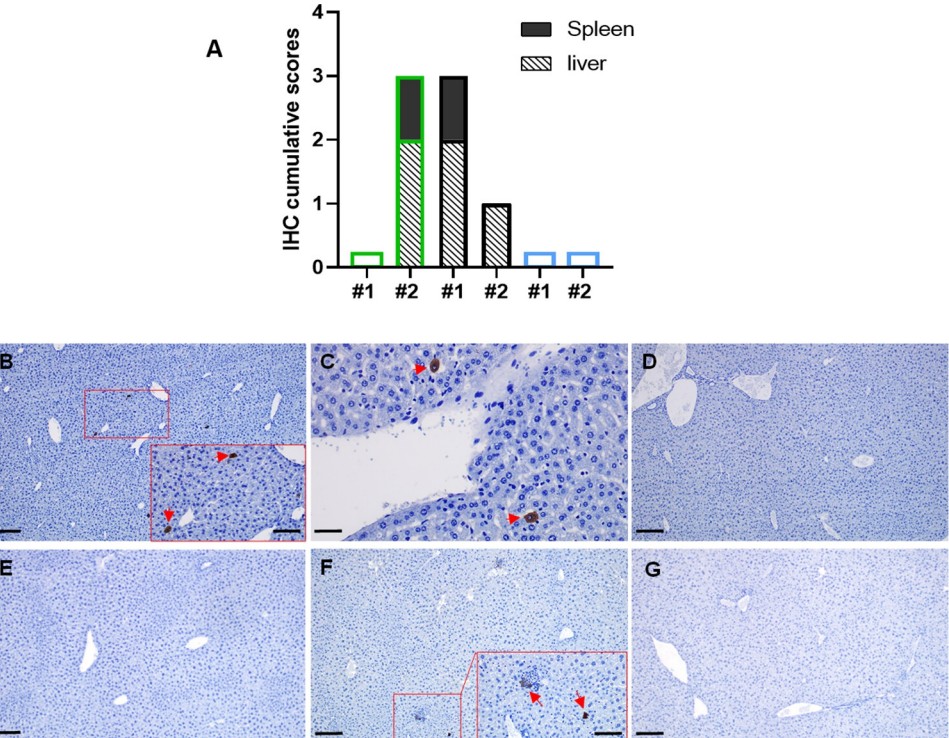

**Fig 4. Histopathological studies on schedule euthanized mice.** 11–15 week old IFNARKO mice, including both males and females, were IN inoculated with 10 microliters of viral suspension containing 1000 pfu of rG1 (n = 2), rMP-12 (n = 2) and 40Fp8 (n = 7). Animals were assessed for signs of disease and survival only, and euthanized on days 2, 5 and 9 pi. Based on previous survival results, groups receiving rG1 and rMP-12 were scheduled only for day 2. Brain and liver samples were collected for virus isolation and RT-qPCR while liver, brain, spleen and kidney samples were fixed in 4% buffered formalin solution for histopathological (HP) and immunohistochemical (IHC) studies. **(a) Individual representation of the cumulative scores** of cells immunolabelled against RVF virus antigen in tissue samples obtained on day 2 pi from mice infected with rG1, rMP-12 and 40Fp8. Immunolabeled cell score (y-axis); Mice evaluated in each virus group (x-axis). The borders of the bars have been coloured according to the virus inoculated: green: rG1; black: rMP-12; blue: 40Fp8**; (b-g) Representative images of liver sections immunolabelled against RVF virus antigen.** Note the occasionally presence of immunolabeled cells, mainly hepatocytes, in the liver of rG1 (b, inset; arrows) and rMP-12 (c; arrows) inoculated mice on day 2 pi, as well as the absence of viral antigen in the liver of 40Fp8 inoculated mice on the same date (d). In mice inoculated with 40Fp8 and euthanized on day 5 pi (e, f), viral antigen was only detected ocassionally in the liver of one mouse (f, inset; arrows). Viral antigen was not present in any of the tissue sections taken from the mice euthanized on day 9 pi (g). (b) Liver, rG1#2 mouse, 2 dpi; (c) Liver, rMP-12#1 mouse, 2 dpi; (d) Liver, 40Fp8#1 mouse, 2 dpi; (e) Liver, 40Fp8#1 mouse, 5 dpi; (f) Liver, 40Fp8#2 mouse, 5 dpi; (g) Liver, 40Fp8#1 mouse, 9 dpi. IHC, scale bars: 50 micrometers (c); 100 micrometers (insets: b,f); 200 micrometers (b,d,e,f,g).

rendered a Cq value of 32.31, with only occasional immunolabelled cells appearing (Fig 4F). No virus antigen was detected by IHC analyses in any other sample from brain or liver, from day 5 or 9, not even in those brain samples rendering Cq values in the limit detection range.

## Discussion

The pathogenesis of virulent RVFV strains in the mouse model is characterised by rapid death of most animals 3–6 days after infection with severe hepatitis. Those that survive this acute onset of liver disease, despite viral clearance from visceral organs, succumb to late-onset encephalitis around day 9–10, with viral neuroinvasion and neurological signs [40–42]. Although this is not the usual pattern of RVFV infection in livestock, which is mainly charac-terised by reproductive failure and hepatic injury [1,43], this dual hepatic-neurological disease

is a typical outcome of severe RVFV infection in humans [2], making studies in mice particularly relevant.

The route of inoculation can lead to different patterns of virus dissemination with differences in the main organs targeted and the kinetics of viremia [44]. The intranasal (IN) route constitutes an extremely severe form of infection in rodents, with earlier and more severe neuropathology, higher mortality rates and higher virus levels in the brain in immunocompetent mice, even after inoculation with attenuated virus. It has been suggested that IN infection favours the entry of the virus into the central nervous system and thus the development of the neuropathologies described in RVF [15,16,45].

In this work we combined two extremely sensitive procedures using of A129 IFNARKO mice, that provide an extremely sensitive system for assessing the degree of viral attenuation [32,33] (for a review, see [46]), and the IN route of administration to demonstrate extreme attenuation of the 40Fp8 RVFV variant. 40Fp8 has been proposed as a safe live attenuated vaccine candidate based on its immunogenicity and safety characteristics. Wild type mice inoculated with 40Fp8 by the intraperitoneal route at doses up to $10^4$ developed a protective immune response, while the same dose administered by the same route in highly sensitive A129 mice behaved as attenuated, causing no signs of disease. [31]. Here we have confirmed its attenuated phenotype even after IN inoculation of 1000 pfu and, furthermore, demonstrated that its degree of attenuation is much higher than that of two viruses representing two live attenuated vaccines prototypes: rMP-12, attenuated by a combination of a few single attenuation determinants [18], and rG1, a recombinant virus lacking the NSs protein, which resembles the attenuation mechanism of the Clone 13 vaccine and its derivatives. While these two viruses caused the death of 100% of the animals within a few days even at the lowest dose tested, 50 pfu, with virus detected in most target organs, the group receiving 1000 pfu of 40Fp8 showed 87.5% (7/8) survival with no signs of disease in most animals, no detectable virus in blood on day 3, and no viral staining or tissue lesions according to histopathological and immunohistochemical studies.

All blood samples taken on day 3 pi from mice inoculated with 40Fp8 were negative by RT-qPCR, indicating a very low level of replication. However, the same result was obtained with some samples from animals inoculated with rG1 and rMP-12, which nevertheless succumbed in the following days. Detection of viral nucleic acids in biological samples (blood, tissue homogenates) by RT-qPCR can sometimes provide discrepancies, with both false negatives and positives due to the presence of impurities in the sample that can interfere with nucleic acid extraction, degrade the RNA or inhibit the polymerase, especially in samples with viral loads close to the sensitivity of the assay [47]. Inoculation of cultured cells allowed a better assessment of positive or negative samples, confirming the absence of infectious virus in blood samples collected on day 3 pi from mice inoculated with 40Fp8, and thus that 40Fp8 viral replication levels or kinetics are not high/fast enough to cause acute lethal disease in most animals.

In any case, seroconversion demonstrated that all survivors were indeed exposed to the virus and supported some level of viral replication, as antibodies to the N protein could be detected. Based on neutralising antibody levels, which are accepted as the clearest correlate of protection, it would be expected that most animals in the 1000 pfu inoculated group would be protected, whereas predicting protection in the remainder is difficult, although we (previous studies by our group) and other studies have already demonstrated protection against RVFV in the absence of neutralizing antibodies, and the involvement of other mechanisms of immunity [48,49].

In contrast to the acute deaths of animals inoculated with rG1 and rMP-12 between day 3 and 5 pi, the only death in the group inoculated with 40Fp8 (experiment 1) was associated to neurological signs observed in one mouse on day 9 pi, which recommended its euthanasia. However, this animal showed no presence of viral antigen in the brain, neither in circulating

cells nor in cells constituting the brain parenchyma, contrary to other studies in which neurological damage late after infection has been correlated with the presence of virus in the brain [15,40,42,50]. Even though virus was not detected in samples from the brain and other organs, severe lymphocytic meningitis was observed (Fig 2I), probably induced by the viral infection, that may have contributed to the neurological signs that forced the euthanasia of this animal. In the absence of virus detection, organ lesions, or blood chemistry data, it is difficult to explain the signs that led to the sacrifice of this animal. The only objective sign of infection found in mice inoculated with 40Fp8 was the presence of virus in liver samples from 40Fp8#2 mouse taken on day 5 in the second experiment, with a pattern of results (isolation of infectious virus, positive Cq in RT-qPCR and very weak staining of viral antigen) quite similar to that of samples taken on day 2 from animals inoculated with rG1 or rMP-12 (Fig 4B, 4C and 4F), which were expected to die in the following days. Perhaps this delay in virus replication and in the onset of liver damage shown by 40Fp8 infection (5 dpi versus 2 dpi) allows the triggering of an immune response capable of preventing acute death of the animal, but not enough to prevent other systemic lesions. Since these observations are from single animals, it is difficult to elucidate the pathogenesis caused by 40Fp8 replication, which would require a more accurate study with daily sampling of a significant number of animals inoculated with higher doses of virus. However, the high attenuation of this virus, even in this susceptible mouse strain, makes it difficult to approach. In our first experiment, only 1 of the 8 animals in the group receiving the high dose of 1000 pfu (12.5%) developed an infection leading to severe illness and death, so that a large number of animals would be required to obtain positive viral samples, which would be a handicap from an animal welfare point of view.

In conclusion, the results described here show that RVFV 40Fp8 is a highly attenuated virus, which even when inoculated intranasally, considered as a severe route of infection, produces high survival rates with no detectable/noticeable tissue damage in an IFNARKO mouse strain that is extremely susceptible to viral infections. Despite its low level of replication, this virus was able to induce immunity in vaccinated animals, thus confirming the potential of 40Fp8 as a safer attenuated vaccine, with a much higher level of attenuation than other live attenuated viruses currently used for livestock vaccination or conditionally accepted as human vaccines. In addition, our results support the classification of RVFV 40Fp8 as Hazard Group 2, as RVFV MP-12, thus allowing its handling in Biosafety Level 2 facilities and facilitating vaccine testing under field conditions.

Rift Valley fever has been identified by the WHO R&D Blueprint as one of the top ten priority pathogens that "pose a public health risk because of their epidemic potential and for which there are no, or insufficient, countermeasures". In addition to the human impact, the repercussions in endemic areas in underdeveloped tropical countries, where livestock is the main source of income and food, are dramatic, causing important monetary losses to producers in both the short and long term. The availability of a vaccine that provides safe immunisation against RVFV, with potential use in both humans and livestock will help to address the problems derived from RVFV infection. Once demonstrated its high attenuation in an immunodeficient model, further work is underway to assess the suitability of 40Fp8 as an RVF vaccine. Studies to determine the stability and yield of 40Fp8 in cell culture, its transmissibility within vaccinated animals as well as its safety and efficacy in susceptible natural hosts such as pregnant sheep are currently in progress.

## Supporting information

**S1 Table. RVFV detection in whole blood samples from day 3 pi.** Groups of n = 8 A129 mice 8–12 weeks old were inoculated by the IN route with a low (L) or high (H) dose (50 or

1000 pfu, respectively) of the indicated viruses: 40Fp8 (S1A Table), rG1 (S1B Table) and rMP-12 (S1C Table). Each group included 4 Female (F) and 4 Male (M). Whole blood samples collected at day 3 pi were processed to test for viral load by RT-qPCR. Samples rendering a Cq value over o close to 33.0 were further assayed for infectious virus isolation (VI) by inoculation of 5–20 microliters (depending on the availability) on cell culture. A few samples were included as internal controls of the assay (M_L_3; F_L_4). Labelling of the animals is the same as in Figs 1C and 2A. Day of death for each animal is indicated. FD: found dead (in some cases blood sample could not be collected); EU: euthanized; S:survivor at day 15 pi (end of the experiment). For RT-qPCR results, the Cq given by the sample is indicated. The results of virus isolation are expressed as follows: NEG (negative, no cpe after 3 passages); (++) (very positive, cpe registered after 1st passage). The specificity of the cpe was confirmed by IFI. ND = not done.
(DOCX)

**S1 Fig. Scheme of the experimental design of the two trials described in the paper for the inoculation and monitoring of mice, and sample collection and analysis. (a) Experiment 1**, virulence assessment; **(b) Experiment 2**, pathological findings.
(TIF)

**S2 Fig. Histopathological studies.** 40Fp8-inoculated mice euthanized on day 15 pi, identified by sex (M/F) and viral dose received (H/L) as in the legend of Fig 1, were selected for histopathological (HP) and immunohistochemical (IHC) studies (mice M_H_2, M_H_3, F_H_1 and F_H_3). Data for each animal are shown in S1A Table. Representative images of tissue sections (liver, spleen, kidney and brain) immunolabelled against RVF virus antigen are shown. Immunolabeled cells were not observed in any of the selected tissues. IHC, black scale bars: 200 micrometers.
(TIF)

**S1 Data. Excel spreadsheet containing, in separate sheets, the underlying numerical for figure panels.**
(XLSX)

# Acknowledgments

We thank Antonia González Guirado and Laura Fernández del Ama for excellent technical assistance.

# Author Contributions

**Conceptualization:** Belén Borrego, Alejandro Brun.

**Formal analysis:** Belén Borrego, Celia Alonso, Pedro José Sánchez-Cordón, Alejandro Brun.

**Funding acquisition:** Belén Borrego, Alejandro Brun.

**Investigation:** Belén Borrego, Celia Alonso, Sandra Moreno, Nuria de la Losa, Pedro José Sánchez-Cordón, Alejandro Brun.

**Methodology:** Belén Borrego, Celia Alonso, Sandra Moreno, Nuria de la Losa, Pedro José Sánchez-Cordón, Alejandro Brun.

**Project administration:** Alejandro Brun.

**Resources:** Belén Borrego, Sandra Moreno.

**Supervision:** Belén Borrego, Alejandro Brun.

**Validation:** Belén Borrego, Celia Alonso, Pedro José Sánchez-Cordón.

**Visualization:** Belén Borrego.

**Writing – original draft:** Belén Borrego, Celia Alonso, Pedro José Sánchez-Cordón.

**Writing – review & editing:** Belén Borrego, Alejandro Brun.

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
