## [Decision Letter · Decision Letter 0]

28 May 2024

Dear Dr. Borrego,

Thank you very much for submitting your manuscript "The Rift Valley fever (RVF) vaccine candidate 40Fp8 shows an extreme attenuation in IFNARKO mice following intranasal inoculation" for consideration at PLOS Neglected Tropical Diseases. As with all papers reviewed by the journal, your manuscript was reviewed by members of the editorial board and by several independent reviewers. In light of the reviews (below this email), we would like to invite the resubmission of a significantly-revised version that takes into account the reviewers' comments. 

We cannot make any decision about publication until we have seen the revised manuscript and your response to the reviewers' comments. Your revised manuscript is also likely to be sent to reviewers for further evaluation.

Sincerely,

Prashant Kumar, Ph.D.

Academic Editor

Mabel Carabali

Section Editor

Reviewer's Responses to Questions

**Key Review Criteria Required for Acceptance?**

**Methods**

-Are the objectives of the study clearly articulated with a clear testable hypothesis stated?

-Is the study design appropriate to address the stated objectives?

-Is the population clearly described and appropriate for the hypothesis being tested?

-Is the sample size sufficient to ensure adequate power to address the hypothesis being tested?

-Were correct statistical analysis used to support conclusions?

-Are there concerns about ethical or regulatory requirements being met?

Reviewer #1: This is a clearly written manuscript on the the Rift Valley fever vaccine candidate 40Fp8 that discusses attenuation in mice. 

1. The objectives of the study are clearly articulated with a testable hypothesis

2. The study design is appropriate and the mice population for testing is clearly described

3. The statistical analyses are limited, beyond stating what software was used and the significance level for p-values, the aim and scope of the statistical tests are lacking

4. I would have liked to see a schematic or table explaining what was observed or tested for on which days. It was difficult to work out what was checked for on each day relying only on written information in text format. For example, what was checked on each specific day: "Day 3: blood samples were taken for analysis of viremia by RT qPCR... Days 6-7: Cultures were kept in the incubator at 37º C and checked up to 6-7 days for cytopathic effect (cpe) fluorescence in the case of virus rG1), etc."

5. citation format is inconsistent throughout, i.e. why is there a website link on lines 66-67?

Reviewer #2: Objectives clearly articulated and hypothesis clearly stated. 

The study design in general was appropriate for the hypothesis tested. 

The power is sufficient and statistical analysis is correct to support conclusions.

Reviewer #3: The methods are well described for all aspects of the study. I would recommend that the ethics statement (Line 171 is placed at the start of the section on "Mice inoculation and sampling" rather than the end.

Minor editing needed for reference citation, line 143.

The statistical analysis is sufficient for the study.

**Results**

-Does the analysis presented match the analysis plan?

-Are the results clearly and completely presented?

-Are the figures (Tables, Images) of sufficient quality for clarity?

Reviewer #1: 1. the figures are clearly presented and captions make sense, I appreciated the explicit text re shaded regions and interpretation of the box and whiskers

2. the figures are of sufficient quality

3. I would have expected more statistical analyses beyond just stating one p-value, this would make the findings of the paper stronger

Reviewer #2: The results are clear and well presented with quality and clarity.

Reviewer #3: The analysis of the data is acceptable

**Conclusions**

-Are the conclusions supported by the data presented?

-Are the limitations of analysis clearly described?

-Do the authors discuss how these data can be helpful to advance our understanding of the topic under study?

-Is public health relevance addressed?

Reviewer #1: 1. The findings of the animal models are clearly summarised, but again would have expected more on the statistical conclusions 

2. There is little or no commentary on the public health relevance of this vaccine, if the vaccine were to demonstrate efficacy in animal models, what would be the next steps? Could this be a vaccine that becomes an intervention in the human population? What would be the limitations or evidence gaps for this vaccine in the human population? What can we learn from other vaccines under development or recently licensed (i.e CHIKV)? What are the considerations for next steps in the vaccine development pathway for RVF?

Reviewer #2: Conclusions support the data. The limitations can be discussed better. The authors do a good job in addressing the public health relevance.

Reviewer #3: The conclusions of the study are supported by the results presented, namely that the attenuated strain of RVFV 40Fp8 is highly attenuated relative to others an minimally pathogenic to interferon receptor ko mice.

The one area of concern is the unexplained death on one animal inoculated with high dose RVFV strain 40Fp8. The authors attribute this to liver dysfunction in the early stages of infection (line 451) leading to "toxic substances in the blood causing neurological signs. This is not very convincing and not supported by the data presented showing minimal virus presence in the liver compared to infection with other RVFV attenuated strains. This needs revising and a more plausible explanation for the death of this animal.

**Editorial and Data Presentation Modifications?**

Reviewer #1: minor revisions, the paper could benefit from more robust statistical analyses and explanation of the public health relevance of these findings

Reviewer #2: The figures with microscopy images can have scale bar.

Reviewer #3: The Discussion in its present form needs to be reduced considerably. Lines 364 to 402 give background information that could be discarded without compromising the manuscript.

Minor edits:

Lines 58-60, revise so that the sentence makes sense.

Line 259, reference to Fig 1c seems inappropriate as this data is not HP or IHC.

Line 260, state precisely how many mice.

Line 410, spelling - demonstrated

Line 425, revise to "..with both false negatives and positive.."

Line 426, spelling - extraction

Line 629, spelling - occasionally

Figure 4 legend, state the colour scheme used.

**Summary and General Comments**

Reviewer #1: (No Response)

Reviewer #2: The present work by Borrego et.al. aimed to extensively test the attenuation of a live attenuated vaccine candidate 40Fp8 against Rift Valley fever (RVF) arbovirus, utilizing an extremely immunodeficient, interferon alpha receptor knockout (IFNARKO) mouse model with an aggressive intranasal inoculation route. The authors performed well thought out experiments by well-established methods to carefully assess the attenuation of 40Fp8 and compared this to previously well characterized and licensed live attenuated virus (LAV) strains having deletion in non-structural segments (NSs-) and another possessing twelve mutations in various genomic segments MP-12. The survival studies revealed, 40Fp8 infected mice at both high (10^3 pfu) and low (50 pfu) doses survived for extended times (MST: day 15) with mild to negligible clinical signs, minimal weight loss (high dose group), undetectable virus by qRT-PCR (Cq: 32-37) and replicating virus by IHC and serial passaging (up to three passages) of tissue harvests from major infection susceptible organs such as liver, brain, spleen, and kidneys providing strong basis for extreme attenuation and outweighing the vaccine effectiveness. In comparison, the other two LAV strains NS- and MP-12 infected mice at both the high (10^3 pfu) and low (50 pfu) doses succumbed to death relatively quickly (MST: day 4/5) characterized by detectable viral replication by qRT-PCR (Cq: 20-25), detectable virus by IHC in lesions and replicating virus by passaging (single passage) from major organ tissue harvests. Taken together these results indicate significant attenuation of 40Fp8, albeit with single incidences of mutifocal meningitis (day 9, experiment 1) without detectable viral antigen and in another experiment the authors recovered infectious virus from liver (day 5, experiment 2). 

The claims by the authors are justified by their associated results with well thought out experiments with appropriate controls. The manuscript is well written and cited relevant literature. 

The authors should address the following major concerns:

1. The manuscript lacks a vital data relating to the lungs of the infected mice given the intranasal route of inoculation. The authors must provide the IHC, qRT-PCR and immunofluorescence labelling of the mice lung tissue from each infection group. 

2. Given the concerns the authors proposed relating to escape of other LAVs given the endemic nature of the virus. The authors are encouraged to provide basis for non-susceptibility of 40Fp8 for escape given the rare yet probable infection of naturally circulating virus in endemic animal cohorts. 

3. Acknowledging the challenges, it will be valuable addition to provide virus challenge studies with the replicative virus to show protective efficacy of the 40Fp8 challenged mice? The authors are encouraged to address why was it not included or is being tested? This data is going to be very helpful in providing even higher confidence associated with the present study results. 

4. Given the incidence of neuropathology’s associated with 40Fp8. The authors are encouraged to provide information on the severity of higher 40Fp8 viral doses (>103 pfu) tested? 

5. The authors are encouraged to address if the experiments with Balb/c mice passively blocking InfA (α-InfA) and infected with 40Fp8 were tested and if tested what were the results? 

6. Given that the virus was generated by serial passaging in the presence of favipiravir. The authors are encouraged to provide their insights into 40Fp8 vaccination in the context of pre/post favipiravir treatment of mice? 

7. Further the authors are encouraged to comment on infection efficiency of 40Fp8 in various cell lines (eg: Vero)? It is clear from the authors previous work that 40Fp8 replication fitness is severed due to three mutations in the L gene. Further, it seems helpful if the infection efficiency of 40Fp8 were similar to its parental RVFV strain as the modified 40Fp8 expressing GFP could be potentially be used as substitute for virus neutralizations in BSL2 setting accelerating the testing / development of vaccine candidates for RVFV. The authors can provide insights into the same. 

Following are a few minor comments:

1. Line 286: “data not shown”. The data can be included in the supplementary for clarity and transparency. 

2. Line 351: immunolabelled.

Reviewer #3: The article reports the infection of INFAR KO mice with RVFV strain 40Fp8 to demonstrate the highly attenuated nature of this strain compared to other attenuated strains. Despite intranasal inoculation in a mouse strain lacking a functional interferon response, this strain was unable to cause disease although at high doses could induce an immune response. As the authors allude in the closing remarks of the manuscript, further studies are being attempted to evaluate strain 40Fp8 in ruminants.

Overall the data supports the authors conclusions although the Discussion should be reduced and a more convincing explanation for one death within the 40Fp8 high dose group is needed.

PLOS authors have the option to publish the peer review history of their article (what does this mean?). If published, this will include your full peer review and any attached files.

Reviewer #1: No

Reviewer #2: No

Reviewer #3: No
---

## [Decision Letter · Decision Letter 1]

2 Aug 2024

Dear Dr. Borrego,

We are pleased to inform you that your manuscript 'The Rift Valley fever (RVF) vaccine candidate 40Fp8 shows an extreme attenuation in IFNARKO mice following intranasal inoculation' has been provisionally accepted for publication in PLOS Neglected Tropical Diseases.

Best regards,

Mabel Carabali, M.D., M.Sc., Ph.D.,

Section Editor

Mabel Carabali

Section Editor

Reviewer's Responses to Questions

**Key Review Criteria Required for Acceptance?**

**Methods**

-Are the objectives of the study clearly articulated with a clear testable hypothesis stated?

-Is the study design appropriate to address the stated objectives?

-Is the population clearly described and appropriate for the hypothesis being tested?

-Is the sample size sufficient to ensure adequate power to address the hypothesis being tested?

-Were correct statistical analysis used to support conclusions?

-Are there concerns about ethical or regulatory requirements being met?

Reviewer #1: The authors have addressed my comments and more information on the statistical analysis has been provided

Reviewer #2: Objectives are clearly stated, and the study design is well thought out. The hypothesis is tested appropriately with sufficient controls and the results were analyzed with proper statistical methods. The authors followed the due regulatory requirements for conducting the study.

**Results**

-Does the analysis presented match the analysis plan?

-Are the results clearly and completely presented?

-Are the figures (Tables, Images) of sufficient quality for clarity?

Reviewer #1: Having supplementary materials on findings by day is helpful. Overall, the analysis is improved after more information on the statistical analysis has been provided

Reviewer #2: The experiments are well designed with appropriate controls and the analysis of the results are clear and completely presented in the form of figures and tables.

**Conclusions**

-Are the conclusions supported by the data presented?

-Are the limitations of analysis clearly described?

-Do the authors discuss how these data can be helpful to advance our understanding of the topic under study?

-Is public health relevance addressed?

Reviewer #1: More information has been provided on the real world need for this vaccine

Reviewer #2: The data of the resulting experiments support the hypothesis and the conclusions drawn from these studies. The authors did a good job in modifying the limitations and making it understandable for a broader audience.

**Editorial and Data Presentation Modifications?**

Reviewer #1: (No Response)

Reviewer #2: Accept.

**Summary and General Comments**

Reviewer #1: Accept

Reviewer #2: The study is conducted by well thought out experiments. The results of subsequent sheep immunization studies and individual virus RNA strand associated attenuation studies will be encouraging and of general interest.

PLOS authors have the option to publish the peer review history of their article (what does this mean?). If published, this will include your full peer review and any attached files.

Reviewer #1: No

Reviewer #2: No

---

## [Editor Report · Acceptance letter]

14 Aug 2024

Dear Dr. Borrego,

We are delighted to inform you that your manuscript, "The Rift Valley fever (RVF) vaccine candidate 40Fp8 shows an extreme attenuation in IFNARKO mice following intranasal inoculation," has been formally accepted for publication in PLOS Neglected Tropical Diseases.

Best regards,

Shaden Kamhawi

co-Editor-in-Chief

Paul Brindley

co-Editor-in-Chief
